# Toward Joint Language Modeling for Speech Units and Text

**Ju-Chieh Chou[1]*, Chung-Ming Chien[1], Wei-Ning Hsu[2], Karen Livescu[1],**
**Arun Babu[2], Alexis Conneau[3]†, Alexei Baevski[4]†, Michael Auli[2]**

[1]Toyota Technological Institute at Chicago [2]Meta AI [3]OpenAI [4]Character AI
jcchou@ttic.edu,wnhsu@meta.com

## Abstract

Speech and text are two major forms of human language. The research community has been focusing on mapping speech to text or vice versa for many years. However, in the field of language modeling, very little effort has been made to model them jointly. In light of this, we explore joint language modeling for speech units and text. Specifically, we compare different speech tokenizers to transform continuous speech signals into discrete units and use different methods to construct mixed speech-text data. We introduce automatic metrics to evaluate how well the joint LM mixes speech and text. We also fine-tune the LM on downstream spoken language understanding (SLU) tasks with different modalities (speech or text) and test its performance to assess the model's learning of shared representations. Our results show that by mixing speech units and text with our proposed mixing techniques, the joint LM improves over a speech-only baseline on SLU tasks and shows zero-shot cross-modal transferability.

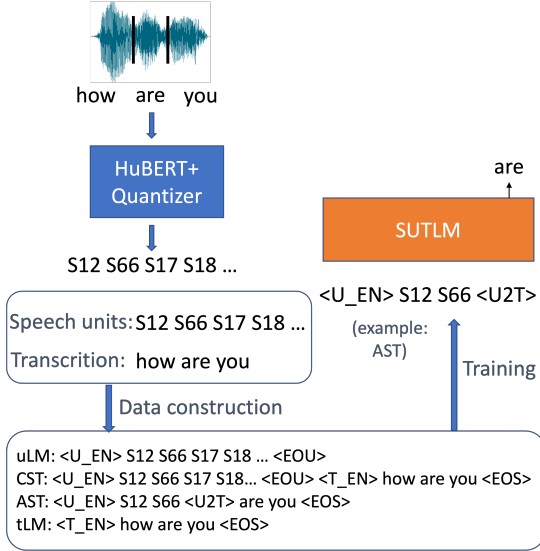

Figure 1: An illustration of our workflow. We tokenize speech signals into discrete units and mix them with text to create speech-text data. Our SUTLM is then trained on a combination of speech-only, text-only, and speech-text data. More details on the data formats can be found in Table 1.

## 1 Introduction

Speech and language processing research has largely focused on spoken and written language separately. However, the integration of speech and text in a single model holds potential benefits. Speech data contains prosodic information that does not exist in the text, which can help in modeling dialogues. On the other hand, text data from sources like Wikipedia can provide structural knowledge that is not available in most speech datasets. Moreover, the amount of written text on the internet exceeds the size of any available speech dataset.

The impressive performance of text large language models (LLMs) has caused a revolution in natural language processing (Radford et al., 2019;

Brown et al., 2020). On the other hand, generative spoken language models (GSLM) (Lakhotia et al., 2021), which are LMs trained on discrete speech units derived from self-supervised representations (Hsu et al., 2021), are also promising for spoken language modeling.

In this work, we aim to fill the gap between text-only and speech-only LMs by developing and studying design choices for a joint Speech Unit and Text Language Model (SUTLM). For speech, we use a self-supervised learning (SSL) speech model, i.e. HuBERT (Hsu et al., 2021), to convert continuous speech signals into speech units. We then combine the units with text data to train an LM that models speech units and text jointly. We convert speech-only, mixed speech-text, and text-only data into token sequences (as shown in Figure 1 and Table 1), and train the model as an LM.

---

*Work done during an internship at Meta AI.
†Work done while at Meta AI.

To evaluate the SUTLM, automatic metrics are developed to quantify the cross-modal ability of the LMs. We also fine-tune our models on downstream tasks for spoken language understanding. We fine-tune the SUTLMs on either the speech or text data and test them on either speech or text to understand how well the models learn to align the two modalities.

Our main contributions are:

- We present a joint autoregressive LM trained on both speech and text (Sec 3).

- We develop automatic metrics that require no fine-tuning for the evaluation of an SUTLM, and show that the proposed metrics are indicative of the model's cross-modal transfer ability on downstream tasks (Sec 4).

- Empirically, we show that units covering a larger span obtained through SentencePiece tokenization (Kudo and Richardson, 2018) outperform local units learned by existing self-supervised models (Hsu et al., 2021) (Sec 5.5.1).

- We find that mixing speech units and text with our proposed techniques (Sec 5.5.3 & Sec 5.5.4) improves the cross-modal ability of the model. (Sec 5.4).

## 2 Related Work

### 2.1 SSL speech models

Self-supervised pre-training enables speech models to learn the information in speech without paired text transcriptions and show impressive performance on tasks such as automatic speech recognition (ASR) with minimal supervised fine-tuning (Baevski et al., 2020; Hsu et al., 2021; Chen et al., 2021). As SSL speech models learn phonetically meaningful speech representations (Pasad et al., 2023), they can be used as a feature extractor (Yang et al., 2021) or a quantizer to transform continuous speech into discrete units (Lakhotia et al., 2021; Lee et al., 2021a,b; Lin et al., 2022; Chen et al., 2022a). In this work, we use the HuBERT model (Hsu et al., 2021) along with a quantizer to tokenize continuous speech into discrete representations. The discrete speech units are then combined with text data to train a single LM that is able to model speech and text jointly.

### 2.2 Textless NLP

Textless NLP (Lakhotia et al., 2021; Polyak et al., 2021; Kharitonov et al., 2021) is a framework to model speech in the absence of textual data. It consists of three components: a speech-to-unit tokenizer, a unit LM (uLM), and a unit-to-speech detokenizer. The tokenizer takes speech signals as inputs to generate discrete speech units. A uLM is trained to predict the next token in an utterance given its prior context. Once the uLM is trained, it can be used to generate unit sequences autoregressively. In the end, the detokenizer is used to convert the generated unit sequences to speech signals.

### 2.3 Joint speech-text transformers

Transformer models have been extremely successful in natural language and speech processing (Vaswani et al., 2017; Gulati et al., 2020), with three major configurations: encoder-decoder models (Vaswani et al., 2017), encoder-only models (Devlin et al., 2018), and decoder-only models (Radford et al., 2018).

Previous works on speech-text joint transformers mostly adapt the encoder-decoder (Ao et al., 2021; Tang et al., 2022; Cheng et al., 2022) or encoder-only (Chung et al., 2020; Bapna et al., 2021; Chen et al., 2022b; Zhang et al., 2022b) architectures. Compared with decoder-only architectures, the training of these models typically requires multiple losses and explicit alignments between paired speech and transcriptions. This makes the hyper-parameter selection time-consuming. Also, encoder-only and encoder-decoder models are mostly used in the pre-training + fine-tuning paradigm, which limits the use cases of these models.

On the other hand, decoder-only models on text (Radford et al., 2019; Brown et al., 2020) show the impressive capability of in-context learning, which also reduces the efforts spent on fine-tuning pre-trained models. In light of this, we explore decoder-only models for speech-text joint training. In this under-explored area, the concurrent work VALL-E (Wang et al., 2023) is the only other attempt to build a decoder-only model jointly modeling speech and text. However, VALL-E's purpose is controllable text-to-speech synthesis (TTS), and the work mainly focuses on the acoustic controllability of the generated speech, while our work aims to build a general-purpose joint LM and mainly focuses on modeling the content of spoken language.

## 3 Method

We start with a dataset of sentences $\mathcal{D} = \{s^1, s^2, \ldots, s^n\}$, where a sentence $s^i$ is composed of a sequence of $T_i$ tokens $(z_1^i, z_2^i, \ldots, z_{T_i}^i)$, where $z_j^i$ can be either text or speech units. The SUTLM is trained to predict the next token $z_j^i$ given its prior context $z_{<j}^i$. We maximize the log-probability of the data

$$\sum_{i=1}^{n} \sum_{j=1}^{T_i} \log P(z_j^i | z_{<j}^i) \qquad (1)$$

In the following sections, we describe how we construct token sequences from speech and text. An example of our data formats can be found in Table 1.

### 3.1 Speech-only: unit LM (uLM)

Prior work has shown that discrete speech units derived from a pre-trained HuBERT model can be used as compact representations to encode speech content, enabling the training of a unit language model (Lakhotia et al., 2021). However, when combining speech with text, the time scales of speech units and text differ. HuBERT units are typically on the phone or sub-phone level, as shown in Table 2. This leads to longer sequences, making it difficult for the model to capture long-term dependencies. On the other hand, subword tokenizers for text generally break text sequences into chunks of a larger size than speech units. This length mismatch between speech and text makes it challenging to model them in a single model. Therefore, we use a subword tokenizer (Kudo and Richardson, 2018) to combine HuBERT units into larger chunks as in (Wu et al., 2022) to mitigate the length mismatch.

The process of generating speech units is as follows. Speech signals are first fed into a HuBERT model. The representations in the final layer are then clustered with the k-means algorithm. The cluster IDs are used as the discrete speech units after removing consecutive repeating units (Lakhotia et al., 2021).[1] These units are then further combined by the subword SentencePiece tokenizer (Kudo and Richardson, 2018). The resulting average number of tokens per second can be found in Table 2.

### 3.2 Text-only: text LM (tLM)

We train another SentencePiece tokenizer (Kudo and Richardson, 2018) using the text-only corpus Sec 5.1.3 to convert text into subword tokens. The resulting vocabulary size of the subword tokens is around 45k.

### 3.3 Concatenated speech-text (CST)

To present paired speech-text data to the SUTLM, we first convert speech units and their transcriptions into the uLM and tLM formats, respectively, and combine them into one sequence by simply concatenating them as shown in Table 1. The CST format explicitly tells the model the correspondence between paired speech and text and thus encourages the model to learn the dependence between speech units and the corresponding text transcriptions.

### 3.4 Alternating speech-text (AST)

Aside from simply concatenating the sequences of speech units and text, we also construct mixed speech-text that takes the word-level correspondence into consideration.

We use a pre-trained speech recognizer (McAuliffe et al., 2017) to force-align speech and its transcription to obtain the word boundaries in an utterance. We then randomly sample some word boundaries within the utterance[2] as the "switching points", which divide the utterance into several chunks. The alternating speech-text (AST) sequence is then constructed by alternatively filling in the chunks with uLM speech units and tLM text tokens, resulting in a sequence that switches modalities at every switching point. Special tokens <U2T> and <T2U> are inserted when switching from speech units to text and text to speech units, respectively.

## 4 Evaluation Metrics

We introduce automatic metrics that require no fine-tuning to evaluate the SUTLM. Fine-tuning is a common approach to assess the quality of pre-trained models (Baevski et al., 2020; Hsu et al., 2021; Chen et al., 2021). However, it is a time-consuming process and the reliability of the experiments highly depends on the hyper-parameter selection process. Furthermore, there is no reliable metric to measure the cross-modal ability of LMs.

---

[1] For example, the unit sequence 13 13 15 80 80 80 becomes 13 15 80 after removing repetitions.

[2] For a sentence with $k$ words, we uniformly sample $\lfloor N \rfloor$ boundaries as the switching points with $N \sim \mathcal{N}(\frac{k}{10}, 1)$.

| Task | Example |
|------|---------|
| uLM | `<U_EN> S12 S66 S17 S18 ... <EOU>` |
| CST | `<U_EN> S12 S66 S17 S18 ... <EOU> <T_EN> how are you <EOS>` |
| CST | `<T_EN> how are you <EOS> <U_EN> S12 S66 S17 S18 ...<EOU>` |
| AST | `<U_EN> S12 S66 <U2T> are you <EOS>` |
| AST | `<T_EN> how <T2U> S17 S18 ... <EOU>` |
| tLM | `<T_EN> how are you <EOS>` |

Table 1: An example of the formats of unpaired (uLM, tLM) and mixed speech-text (CST, AST) data. For the CST and AST formats, speech units and text can be present in a sequence in different orders . `<U_EN>` and `<T_EN>` are used at the beginning of the unit/text sequence. `<EOU>` and `<EOS>` are used at the end of the unit/text sequences. `<U2T>` and `<T2U>` are used when switching from unit to text and text to unit at word boundaries.

|  | Average tokens per second |
|--|---------------------------|
| Phone | 20.32 |
| HuBERT | 50.00 |
| + deduplication | 33.33 |
| + SP 10k | 17.67 |
| + SP 32k | 14.33 |

Table 2: The average number of tokens per second for different types of speech units. SP 10k and 32k refer to SentencePiece tokenization (Kudo and Richardson, 2018) applied to HuBERT units to create a dictionary with 10k and 32k tokens respectively.

In light of this, we propose Context Retrieval Accuracy (CRA), a new metric that does not require fine-tuning, to evaluate the cross-modal ability of an SUTLM.

### 4.1 Context Retrieval Accuracy (CRA)

The motivation of Context Retrieval Accuracy (CRA) comes from the intuition that a good LM should learn to predict the next token based on its prior context. When we divide a sentence into prompt and continuation, a good LM should be able to capture the dependence between them. That is, it should assign a higher conditional probability to the continuation given its corresponding prompt than given a random prompt.

To measure CRA, we gather a collection of $m$ sentences $\mathcal{C} = \{s^1, s^2, \ldots, s^m\}$ and break $s^i$ into a pair of prompt $x^i$ and continuation $y^i$. Given an SUTLM parameterized by $\theta$, we can measure the conditional probabilities $P_\theta(y^i|x^i)$ with Eq 1. The CRA is then computed as:

$$\frac{1}{m} \sum_{i=1}^{m} \mathbb{1}[\arg \max_{j \in \{1...m\}} P_\theta(y^i|x^j) = i], \quad (2)$$

That is, the LM is used as a scorer to classify whether the matched prompt-continuation pair has the highest conditional probability among a pool of unmatched prompts.

CRA also has a pointwise mutual information (PMI) interpretation:

$$\arg \max_{j \in \{1...m\}} P_\theta(y^i|x^j) = i$$
$$\implies \log P_\theta(y^i|x^i) \geq \max_{j \in \{1...m\}} \log P_\theta(y^i|x^j)$$
$$\implies \log \frac{P_\theta(y^i|x^i)}{P_\theta(y^i)} \geq \max_{j \in \{1...m\}} \log \frac{P_\theta(y^i|x^j)}{P_\theta(y^i)}$$
$$\implies \mathrm{PMI}(x^i, y^i) \geq \max_{j \in \{1...m\}} \mathrm{PMI}(x^j, y^i)$$
$$(3)$$

That is, correctly identifying the prompt implies the matched prompt-continuation pair has a higher PMI than all unmatched prompt-continuation pairs.

Ideally, the model should produce similar representations given the same content regardless of the modality. Hence, in addition to the uni-modal CRA, we also consider cross-modal CRA, where the prompt and the continuation are in different modalities. In practice, for example, when we use text as the prompts and speech units as the continuations, we set the probability of emitting text tokens to zero and re-normalize the probability to ensure that the continuation $y^i$ can be only speech units. Cross-modal CRA can be used as a way to measure whether the SUTLM successfully learns shared representations between text and speech.

### 4.2 Perplexity under External LM (PELM)

Following previous work, we use the perplexity under external LM (PELM) to measure the quality of the content of generated samples (Lakhotia et al., 2021). We sample a continuation from the SUTLM given each ground truth prompt. We then use an external text LM, OPT-6.7B (Zhang et al., 2022a),

to compute the perplexity of the sequence:

$$\hat{y}^i \sim P_\theta(y|x^i)$$
$$x'^i, y'^i = \mathrm{T}(x^i \parallel \hat{y}^i)$$
$$\mathrm{PELM}(\theta) = 2^{\dfrac{-\sum_i \log P_{\mathrm{OPT}}(y'^i \mid \mathrm{gt}(x^i))}{\sum_i \mathrm{len}(y'^i)}} \quad (4)$$

where $x^i$ and $\hat{y}^i$ refer to the prompt and sampled continuation, and $\theta$ are the parameters of the SUTLM. Similarly to cross-modal CRA, we control the modality of sampled continuations by zeroing out the probability of the tokens in the undesired modality. Since the prompt and the continuation can be either speech units or subword text tokens, we use a transcriber $\mathrm{T}(\cdot)$ to transcribe the concatenated sequences $x^i \parallel \hat{y}^i$ into text $x'^i, y'^i$.[3] $\mathrm{gt}(\cdot)$ is a function that outputs a ground truth transcription when the input is speech units and is an identity function when the input is text. The external LM is then used to measure the perplexity of the continuation part of the text sequence.

### 4.3 Evaluation on SLUE tasks

We use the SLUE benchmark (Shon et al., 2022) to evaluate our models on downstream tasks. The benchmark includes two tasks, sentiment analysis (SLUE-SA) and named entity recognition (SLUE-NER), with both speech data and transcriptions provided. After pre-training the SUTLM, we fine-tune it on the SLUE dataset with either speech or text data as inputs to predict the ground-truth labels, and then evaluate it on either speech or text inputs. We evaluate the model on different input modalities to understand the cross-modal ability of the model as in (Hsu and Shi, 2022; Bapna et al., 2021, 2022). Fine-tuning details can be found in 5.4.2.

## 5 Experiments

### 5.1 Data

#### 5.1.1 Speech-only

We use 5% of the dataset used in (Aghajanyan et al., 2023) to match the size of the mixed speech-text and text-only data. The dataset includes Multilingual LibriSpeech (MLS) (Pratap et al., 2020), VoxPopuli (Wang et al., 2021), Common-Voice (Ardila et al., 2019) and Spotify Podcast &

People's Speech (Aghajanyan et al., 2023). The subsampled dataset consists of 65k hours of speech.

#### 5.1.2 Mixed speech-text (CST and AST)

We use MLS (Pratap et al., 2020) and VoxPopuli (Wang et al., 2021) to create mixed speech-text data without subsampling. The dataset contains 45k hours of speech and 2.7B of words.

#### 5.1.3 Text-only

We combine OPT web data (Zhang et al., 2022a), Wikipedia, and LibriLM (Panayotov et al., 2015), and then subsample 5% of it, resulting in a total of 8.5B subwords.

### 5.2 SSL speech tokenizer

We use a HuBERT Base model trained on 221K hours of unlabeled speech in 8 languages as in (Hsu et al., 2022; Nguyen et al., 2023).[4] After pre-training, the representations at the last layer (12th) are clustered with k-means using 2000 clusters.

### 5.3 Model architecture and training

We use the 24-layer transformer implementation in fairseq (Ott et al., 2019) with 16 attention heads. The embedding size is 1024, the feed-forward dimension is 4096, and the dropout probability is set to 0.1. The weights of the embedding layer are tied to the output layer (Press and Wolf, 2016). The model contains 350M parameters.

The model is trained for 500k updates on 32 V100 GPUs with a batch size of 8192 tokens per GPU. We use Adam optimizer (Kingma and Ba, 2014) with $(\beta_1, \beta_2) = (0.9, 0.95)$. Gradient clipping with a threshold 1.0 and weight decay of 0.1 are applied to stabilize the training. Since the data size is different for different data formats, we resample speech-only, speech-text, and text-only data equally (1/3 for each in every training batch) to prevent the model from being biased toward any of them.

### 5.4 Evaluation setup

#### 5.4.1 Automatic Metrics

We use a subset of the Multilingual LibriSpeech (Pratap et al., 2020) dev set to evaluate the SUTLM. To provide enough context to the SUTLM, we filter out sentences of less than 20 words. For each sentence, we use the first 10 words as the prompt and the rest as continuation. For the CRA experiments, we evaluate the SUTLM with

---

[3]For both speech units and text tokens, we first invert the SentencePiece tokenization process to get raw HuBERT units and raw text. For speech units, we further use a 12-layer Transformer encoder with a CTC head to map HuBERT units to text. The transformer is trained on LibriSpeech, with a WER of 5.18% on dev-clean, and 11.61% on dev-other.

[4]https://dl.fbaipublicfiles.com/hubert/mhubert_base_vp_mls_cv_8lang_it3.pt

the 100 shortest utterances in the filtered dataset, while for the PELM experiments, we use the 500 shortest utterances. We use fewer utterances in CRA experiments as the computation of CRA is $O(N^2)$ for $N$ utterances. We constrain ourselves to sentences with moderate lengths because the continuation part becomes less coherent with the prompt as the sequence length grows, which hurts the sensitivity of the proposed metrics.

When sampling the speech or text continuations in the PELM experiments, we use temperature $t = 0.6$ and nucleus sampling (Holtzman et al., 2019) with $p = 0.95$, and truncate the continuation to 10 words (identical to the length of the prompts).

### 5.4.2 Downstream Tasks

For SLUE-SA, we fine-tune SUTLM by adding a self-attention pooling layer on top of the transformer model after removing the last output layer (Shon et al., 2022). We fine-tune it with a learning rate of 3e-5 for 30k updates and evaluate it with Macro F1 (Shon et al., 2022).

For SLUE-NER, we follow the SLUE official baseline to formulate the task as an ASR problem and train our model to decode special tokens around each named entity (Shon et al., 2022). We concatenate the output (the text transcription with special tokens before and after each named entity) after the input (speech units when fine-tuned on speech, text tokens when fine-tuned on text) and fine-tune our SUTLM as an LM with the same loss function as Eq 1. The loss is only applied to the output part of the sequence. We fine-tune the SUTLM with a learning rate of 3e-5 for 50k updates. During decoding, we use a beam size of 5 to generate the outputs and evaluate them with Micro F1 (Shon et al., 2022). For both SLUE tasks, we report results on the dev set since the test set is not publicly available. We use the fine-tuned HuBERT as the baseline as in (Shon et al., 2022).

### 5.5 Results

#### 5.5.1 What kind of speech units works the best?

We utilize HuBERT units described in Sec 5.2 (2000 units) and apply SentencePiece tokenizers on them. Results can be found in rows **(A)**, **(B)**, **(C)** in Table 3 for automatic metrics, Table 4 for SLUE-SA and Table 5 for SLUE-NER.

The model trained with SP 10k has the best performance in terms of PELM, SLUE-SA, and SLUE-NER, but slightly worse CRA than the model using

the original HuBERT units. For CRA for the u2u case (unit prompt, unit continuation), we hypothesize that the model uses low-level acoustic information to make predictions as the CRAs are nearly 1.0 for all types of speech units. Also, HuBERT uses overlapping windows for neighboring tokens, so the first token of the continuation contains information about the previous token.

For the speech continuation (PELM) experiments, the SP 10k-based sequences are shorter than HuBERT unit-based sequences, so the model trained with SP 10k (row **(B)**) can generate more coherent continuations.

#### 5.5.2 Do we need paired data to learn shared representations?

In this section, we compare models trained with and without paired data to investigate the usefulness of paired data. We can compare the results in row **(D)** and **(F)** in Table 3 for automatic metrics, Table 4 for SLUE-SA and Table 5 for SLUE-NER. For cross-modal cases (u2t and t2u), in terms of automatic metrics, the model trained with unpaired data alone (row **(D)**) has almost random CRAs and high PELMs, indicating a lack of cross-modal ability.

Similarly, for SLUE-SA, the model trained with unpaired data alone (row **(D)**) shows almost random macro F1 scores for a 3-way classification task when tested on the other modality. For SLUE-NER, the model trained without exposure to paired data (row **(D)**) performs worse than models trained with paired data (row **(F)**) when fine-tuned on speech and shows no transferability between modalities. Row **(D)** also performs worse than its speech unit-only counterpart (row **(B)**, showing that the model trained solely on unpaired data does not demonstrate any cross-modal transfer ability between speech and text.

#### 5.5.3 Does concatenated speech-text (CST) help learn shared representations?

The next question we want to answer is whether CST is helpful in learning shared representations. Building on the previous findings (rows **(A)**, **(B)**, **(C)**), we utilize SP 10k as our speech unit vocabulary and present the results in row **(E)** in Table 3 for automatic metrics, Table 4 for SLUE-SA, and Table 5 for SLUE-NER. The results show that, compared to using unpaired data alone (row **(D)**), the model trained with CST (row **(E)**) has higher CRAs for u2t and t2u, which indicates that the model cap-

|  |  |  |  |  |  | u2u |  | t2u |  | u2t |  | t2t |  |
| row | unit | uLM | CST | AST | tLM | CRA | PELM | CRA | PELM | CRA | PELM | CRA | PELM |
|---|---|---|---|---|---|---|---|---|---|---|---|---|---|
| Ground truth continuation |  |  |  |  |  | - | - | - | - | - | - | - | 101.4 |
| **(A)** | HuBERT | v |  |  |  | 1.00 | 193.3 | - | - | - | - | - | - |
| **(B)** | SP 10k | v |  |  |  | 0.96 | 163.6 | - | - | - | - | - | - |
| **(C)** | SP 32k | v |  |  |  | 0.96 | 177.4 | - | - | - | - | - | - |
| **(D)** | SP 10k | v |  |  | v | 0.94 | 175.9 | 0.03 | 394.9 | 0.01 | 1973.3 | 0.20** | 20.7** |
| **(E)** | SP 10k | v | v |  |  | 0.95 | 166.0 | 0.37 | 39.1* | 0.26 | 43.4* | 0.56 | 34.7 |
| **(F)** | SP 10k | v | v | v | v | 0.97 | 162.8 | 0.70 | 124.7 | 0.81 | 38.7 | 0.67 | 28.2 |

Table 3: Automatic metrics (CRA and PELM). "u2t" denotes that the prompts are speech units and the continuations are text, and so on. (*): for cross-modal cases (u2t and t2u) in row **(E)**, the PELM is low because the continuation simply repeats the prompt. We discuss this issue in Sec 5.6. (**): The low CRA for t2t is due to the use of MLS as an evaluation set, resulting in a distribution mismatch from the text-only training data. Similarly, the use of OPT data to train the SUTLM results in better PELM on t2t in row (D).

|  |  | FT data | SP |  | TXT |  |
| row | unit | Eval set | SP | TXT | SP | TXT |
|---|---|---|---|---|---|---|
| Baseline |  |  | 0.46 | - | - | - |
| **(A)** | HuBERT | uLM | 0.51 | - | - | - |
| **(B)** | SP 10k | uLM | 0.56 | - | - | - |
| **(C)** | SP 32k | uLM | 0.54 | - | - | - |
| **(D)** | SP 10k | uLM+tLM | 0.52 | 0.33 | 0.35 | 0.49 |
| **(E)** | SP 10k | uLM+CST | 0.48 | 0.42 | 0.51 | 0.52 |
| **(F)** | SP 10k | uLM+CST +AST+tLM | 0.49 | 0.43 | 0.52 | 0.56 |

Table 4: Macro F1 score on SLUE-SA. FT data indicates the model is fine-tuned on speech (SP) or text (TXT). Eval set denotes the fine-tuned model is tested on speech (SP) or text (TXT).

|  |  | FT data | SP |  | TXT |  |
| row | unit | Eval set | SP | TXT | SP | TXT |
|---|---|---|---|---|---|---|
| Baseline |  |  | 54.5 | - | - | - |
| **(A)** | HuBERT | uLM | 62.9 | - | - | - |
| **(B)** | SP 10k | uLM | 64.4 | - | - | - |
| **(C)** | SP 32k | uLM | 62.5 | - | - | - |
| **(D)** | SP 10k | uLM+tLM | 63.2 | 1.5 | 0.0 | 66.8 |
| **(E)** | SP 10k | uLM+CST | 65.0 | 3.6 | 0.5 | 79.5 |
| **(F)** | SP 10k | uLM+CST +AST+tLM | 66.6 | 25.2 | 0.3 | 77.2 |

Table 5: The F1(%) score on SLUE-NER. FT data indicates the model is fine-tuned on speech (SP) or text (TXT). Eval set denotes the fine-tuned model is tested on speech (SP) or text (TXT).

tures the relationship between speech and text better than models trained with unpaired data alone.

For SLUE-SA, the model pre-trained with CST shows comparable performance when fine-tuned on one modality and evaluated on the other. The performance when fine-tuning on text and testing on speech is even better than directly fine-tuning on speech (0.51 vs. 0.48). The reason is likely to be that text data provides a less noisy supervisory signal compared to using speech units. The model trained with extra speech-text data (row **(E)**) performs worse than the model trained with only speech units (row **(B)**). The reason may be similar to the "curse of multilinguality" (Conneau et al., 2019), where sharing the capacity of the model with other languages or modalities hurts performance.

For SLUE-NER, concatenated speech-text improves performance over the model trained with only speech units (row **(B)**) when fine-tuned on speech. Unlike SLUE-SA, which is a classification task, here we need to generate the corresponding transcription along with the named entity tags for SLUE-NER. Hence, the model (row **(E)**) fine-tuned on speech benefits directly from the extra speech-text data. We discuss the implications of the fine-tuning results further in Sec 5.7.

For speech / text continuation, when only using concatenated speech-text data (CST) as our mixed data, there are no special tokens (<U2T>, <T2U>) to trigger modality switching. As shown in Table 6, in the u2t case the model trained with CST simply transcribes the speech prompt into its transcription on u2t and synthesizes the text prompt into speech units, resulting in low PELMs for u2t and t2u in row **(D)** due to the repetition. PELM fails to reflect the quality of the continuation accurately. We discuss this limitation further in Sec 5.6.

### 5.5.4 Does alternating speech-text (AST) help learn shared representations?

This section discusses the benefits of alternating speech-text (AST). The results are presented in (row **(F)**) in Table 3 for automatic metrics, Table 4

for SLUE-SA, and Table 5 for SLUE-NER.

By comparing the results of CRA for t2u and u2t in row **(F)** with those in row **(E)** in Table 3, we observe an improvement in CRA when the data is directly constructed to switch modalities on word boundaries. We can also see that CRA is similar for t2u, u2t, and t2t. It suggests that the model learns to match context regardless of modality.

In row **(F)**, PELM for t2u is lower than PELM for u2u as the text prompt is less noisy than speech units. PELM for u2t is only marginally worse than t2t. This shows that the LM trained with AST can continue a sentence regardless of the modality. The worse PELM for u2u and t2u than for u2t and t2t could be attributed to the recognition errors within our unit transcriber.

Regarding SLUE-SA, we can observe that AST and tLM further improve the cross-modal transfer performance (trained on the text and evaluated on speech, or vice versa) in row **(F)**.

In SLUE-NER, row **(F)** also shows better performance than row **(E)** when fine-tuned on speech and evaluated on speech. There is also non-trivial speech-to-text transfer (fine-tuned on speech and evaluated on text) in row **(F)**, showing that AST helps in learning transferable features between modalities.

In SLUE-NER, when fine-tuned on text and evaluated on speech, there is no transferability between speech and text. The reason can be attributed to the fine-tuning task becoming almost trivial. In text NER, in our formulation, the input and output are nearly identical. The only difference is the named entity tags. Further discussion of downstream task performance can be found in Sec 5.7.

### 5.6 Limitations of PELM

We use PELM as a metric to measure the quality of continuations. However, although our SUTLM (row **(F)**) shows the ability to continue after a cross-modal prompt, the resulting continuation is still only locally consistent as shown in Table 6. This can be attributed to the use of a 350M-parameter model architecture, which is relatively small in the era of LLMs.

The PELM metric fails to accurately reflect the result in the case of row **(E)** when the model simply repeats the prompt. It has been a known phenomenon that LMs tend to assign a high probability to repeated tokens (Holtzman et al., 2019).

To quantify repetition, we compute the propor-

tion of bi-grams in continuations that have appeared in the prompt transcription. For row **(E)**, the proportions are 0.02, 0.53, 0.42, and 0.02 for u2u, u2t, t2u, and t2t, respectively. For row **(F)**, the proportions are 0.02, 0.03, 0.01, and 0.03. For row **(E)**, the continuations for u2t and t2u are simply repeating the content of the prompt.

We can see that the u2t and t2t PELMs are lower than the ground truth PELM. This is because of the use of the temperature of 0.6 in the softmax layer, which likely hurts diversity and coherence as in (Caccia et al., 2018; Lakhotia et al., 2021).

### 5.7 Implications for SLU Downstream Tasks

We show that mixing speech units and text improves the cross-modal ability of the model. In SLUE-SA, the mixed speech-text data enables the model to zero-shot transfer between speech and text. In SLUE-SA, we remove the output layer from the SUTLM and attach a classification head so the model will always output a valid class.

In SLUE-NER, using mixed speech-text data directly improves the performance. Since this is a sequence generation task, the mixed speech-text data helps the model generate better text. The transfer from speech to text is non-trivial but not vice versa. This finding aligns with the experiments in (Bapna et al., 2022), in which they also find non-trivial transfer from speech to text but not the other way around. However, we note that different fine-tuning strategies can produce different results, as demonstrated in (Liu et al., 2021).

## 6 Conclusion

Our study on joint language modeling for speech units and text involved developing evaluation metrics and fine-tuning the model on speech and text data. We found that using mixed speech-text data improves the model's cross-modal ability and performance on both automatic metrics and downstream tasks.

Our study sheds light on the benefits of considering both speech and text in building language models. We hope that this research will motivate the research community to further explore the integration of speech and text data for more comprehensive language modeling.

Future work in this area could involve investigating the optimal balance between speech and text data in model training and exploring ways to handle multi-modal data beyond the speech-text domain.

# 7 Limitations

Our approach involves using a speech tokenizer that can encode phonetic information (HuBERT) and an off-the-shelf speech recognizer to generate word-level alignment. For other, lower-resource languages, these components may be harder to obtain or may not perform as well.

For our proposed automatic metrics, the complexity of CRA grows at a rate of $O(N^2)$, which can be expensive when evaluated on a larger number of utterances or when scaling up the model size. PELM, on the other hand, also has limitations as stated in Sec 5.6. For the empirical results on downstream tasks, we test our SUTLMs on the SLUE benchmark, which has only two tasks. Extending the experiments to more downstream tasks may provide more insights.

Finally, we only study relatively small SUTLMs (350M parameters). It is unclear how scaling it up would affect the results.

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

## A  Appendix

| prompt | they need the same aid and they | |
|---|---|---|
| **model** | **modalities** | **Continuation** |
| **(E)** | u2u | soon will increase and reach the grave more gradually than ever they seek their own home and many little people suffer from the hospitality and care of the poor |
| | u2t | they need the same aid and they need the same aid |
| | t2u | they need they they need the sameling the need the need the same sense of will that they need for others others to to to to do they will a and and they to have |
| | t2t | need the same help they can make and are able to make the best of them and the best of them |
| **(F)** | u2u | certainly do need the same money and they do not give themselves out of either field or settlement but they do need it they say and they need it |
| | u2t | shall be able to perform it all the same the king of egypt is not free from the persecution of the jews he is free from the persecution of the jews |
| | t2u | can no longer face it but though these men believe women now to be the principal duties the progress made will actually continue |
| | t2t | will be the same as they have been when they were born and they will be the same as they now are if they were not born they would be the same as they now are |

| prompt | barrack was just coming through the doorway carrying half | |
|---|---|---|
| **model** | **modalities** | **Continuation** |
| **(E)** | u2u | doesn papa and several small scotch girls who had been sitting by the fire with him he said to the girls on whom he kept his second mother mother is good to me and so i would not let her go away |
| | u2t | barrek was just coming through the doorway carrying half a dozen other things |
| | t2u | round of the door he was just coming through the doorway carrying half around of the door they stopped at the back of the doorway and half a around of the doorway and half a round of the doorway |
| | t2t | a dozen men and a dozen men and women i believe and i think i could have saved the lives of those who loved me-but i don't know how i could have saved them |
| **(F)** | u2u | dozen boxes when he saw the stick black inside in his room stepping out of his way into the hall a chuckle of joy fell in the drawing room and he seized the boxes and broke down the door |
| | u2t | a cup of coffee when gertie came up with a basketful of rice and a handful of water and then she came to the house of mrs smiths and she said to gertie |
| | t2u | dozen packs and a wrapper and a light sparkling light across the face of jack and a burning gold bullet and a very sharp thumb |
| | t2t | a dozen toy guns and a hundred toy guns and a hundred toy guns |

Table 6: Example for speech and text continuation. Speech continuation has been transcribed by the transcriber.