# OpenReview forum: "Toward Joint Language Modeling for Speech Units and Text"
_EMNLP/2023/Conference — EMNLP 2023 Findings_

### Official Review · Reviewer_oo6z · 2023-08-04

**Soundness:** 3

**Excitement:**

3: Ambivalent: It has merits (e.g., it reports state-of-the-art results, the idea is nice), but there are key weaknesses (e.g., it describes incremental work), and it can significantly benefit from another round of revision. However, I won't object to accepting it if my co-reviewers champion it.

**Paper Topic And Main Contributions:**

The article focuses on joint modeling of speech units and text, empirically validating that a larger span tokenizer achieves better modeling results. Additionally, the authors propose an automatic evaluation metric that does not require fine-tuning to assess the model’s cross-modal transfer capability.

**Reasons To Accept:**

The writing in the paper is clear, and the authors introduce a new evaluation metric. Also, experiments comparing the effectiveness of different levels of speech tokenizers and various methods for mixing speech-text data are conducted.

**Reasons To Reject:**

In works such as [AudioLM](https://arxiv.org/pdf/2209.03143.pdf) and [SpearTTS](https://arxiv.org/pdf/2302.03540.pdf), HuBERT is understood to be biased towards semantic rather than acoustic tokens. The proposed CRA metric and downstream tasks such as sentiment analysis, NER, and ASR also seem to focus more on language aspects. This paper may be more focused on modeling text rather than jointly modeling speech and text. However, the paper emphasizes the model’s cross-modal capability multiple times. To further support the effectiveness of the model in speech-text joint modeling, it would be beneficial to consider adding tasks related to speech, such as speech reconstruction.

**Reproducibility:**

4: Could mostly reproduce the results, but there may be some variation because of sample variance or minor variations in their interpretation of the protocol or method.

**Reviewer Confidence:**

2: Willing to defend my evaluation, but it is fairly likely that I missed some details, didn't understand some central points, or can't be sure about the novelty of the work.

---

> ### Author Rebuttal · Authors · 2023-08-29
>
> We thank the reviewer for the comments.
>
> In this work, we mainly study tasks related to semantic understanding, which is more likely to benefit from the text. For more acoustic-level speech generation tasks, one may consider extending the joint LM to finer-grained tokens such as in AudioLM, or adding a vocoder to synthesize speech from Hubert units.

---

### Official Review · Reviewer_HZX6 · 2023-08-05

**Soundness:** 3

**Excitement:**

2: Mediocre: This paper makes marginal contributions (vs non-contemporaneous work), so I would rather not see it in the conference.

**Missing References:**

For CRA y'all can cite this work which does cross-modal retrieval:

@inproceedings{conneau2023fleurs,
  title={Fleurs: Few-shot learning evaluation of universal representations of speech},
  author={Conneau, Alexis and Ma, Min and Khanuja, Simran and Zhang, Yu and Axelrod, Vera and Dalmia, Siddharth and Riesa, Jason and Rivera, Clara and Bapna, Ankur},
  booktitle={2022 IEEE Spoken Language Technology Workshop (SLT)},
  pages={798--805},
  year={2023},
  organization={IEEE}
}

**Paper Topic And Main Contributions:**

This paper identifies a gap in the joint modeling of speech and text. Prior works have focused on encoder-only or encoder-decoder architectures while this work proposes to build a multimodal (speech-text) decoder. The decoder is trained with four objectives: a) unitLM (uLM) : maximizes the log-probability of discretized speech-only data (speech units obtained using the pre-trained HuBERT model); b) textLM (tLM): maximizes the log-probability of text-only data; c) concatenated speech-text (CST): maximizes the log-probability of discretized speech and its text transcript concatenated together; and d) alternating speech text (AST): mixed speech-text data is constructed by switching between the two in a single utterance. The paper additionally proposes an evaluation metric for evaluating speech-text models without fine-tuning, called "context retrieval accuracy" (CRA), which aims to measure whether a model can retrieve the correct continuation for a given prompt, in a cross-modal way too. Evaluation is done on SLUE sentiment analysis and named entity recognition.

**Questions For The Authors:**

A) What is the baseline model mentioned in Tables 4 & 5? I couldn't find a description of the baseline system in the text.

B) Any idea on how the model performs for text-only tasks and generation-tasks? (if y'all have additional experiments that didn't make it to the paper)


**Reasons To Accept:**

A) Analysis: Detailed analysis of the contribution of each pre-training objective: uLM, tLM, CST and AST. The paper goes beyond simply stating results and analyses why they may be so.

B) Writing: Clear writing that is easy to understand. Situates the work well in the related-work section.

**Reasons To Reject:**

A) CRA metric: The CRA metric tests what the AST metric is optimizing. While I see value in cross-modal retrieval, i.e., retrieving a relevant speech sample for a given text sample and vice-versa (for example, in audio mining for SSL), I am not sure why intra-sentential cross-modal completion is relevant. Cross-modal retrieval has already been addressed by past works for example in [Conneau et. al., 2022](https://arxiv.org/abs/2205.12446). It is also not clear how CRA correlates with downstream task performance for it to be advocated as a replacement for fine-tuning (as mentioned in L070) because only the final model (Row F) can be used to measure CRA across all 4 directions.

B) Experimental Setup: The baselines used are not mentioned (as far as I could search). Since sentiment analysis and NER are classification tasks, the paper can include encoder or encoder-decoder models as baselines as well (like [SLAM](https://arxiv.org/pdf/2202.01374.pdf)?). Further, since this is a multimodal decoder-only model, I was expecting evaluation on generation tasks like ASR and text-only tasks (like NLI or NER) to fully grasp the capabilities of the model. That said, I completely understand if it was out of scope for the current work given resource constraints. In that case, it may be helpful to lay out the scope as such.

**Reproducibility:**

4: Could mostly reproduce the results, but there may be some variation because of sample variance or minor variations in their interpretation of the protocol or method.

**Reviewer Confidence:**

3: Pretty sure, but there's a chance I missed something. Although I have a good feel for this area in general, I did not carefully check the paper's details, e.g., the math, experimental design, or novelty.

**Typos Grammar Style And Presentation Improvements:**

-

---

> ### Author Rebuttal · Authors · 2023-08-29
>
> We thank the reviewer for the comments.
>
> Re reject reason (A):
> Similar to [Conneau et. al., 2022](https://arxiv.org/abs/2205.12446), we also use multi-modal retrieval as a way to probe the model. Instead of using a fixed-size vector in an encoder-only model, we modify the retrieval target to the context to fit the auto-regressive nature of LM. It also has a PMI interpretation as shown in eq(3). While only row (F) is explicitly trained to continue the sentence, row (E) also shows a non-trivial CRA and non-trivial transferability at the same time.
>
> Re reject reason (B):
> NER is not a classification task, since it involves locating and labeling an arbitrary number of NEs.  As in prior work on spoken NER, we formulated the task as a generation task, as described in L405-409. It would be great if we could compare it to SLAM. However, SLAM is not open sourced so we won’t be able to directly compare to it using the same setup.
>
> Re question (A):
> The baseline is a fine-tuned HuBERT from [Shon et.al.](https://arxiv.org/pdf/2111.10367.pdf). We will add the reference to the final version.
>
> Re question (B):
> In the SLUE-NER experiment, when fine-tuning on text and testing on text, it can be considered a text-only / generation task. Also, we would like to point out that the NER task is very similar to ASR, but we only evaluate the named entity spans.

---

### Official Review · Reviewer_rsNT · 2023-08-10

**Typos Grammar Style And Presentation Improvements:** Line #273
**Soundness:** 3

**Excitement:**

2: Mediocre: This paper makes marginal contributions (vs non-contemporaneous work), so I would rather not see it in the conference.

**Missing References:**

Zhang, Ziqiang, et al. "SpeechUT: Bridging Speech and Text with Hidden-Unit for Encoder-Decoder Based Speech-Text Pre-training." Proceedings of the 2022 Conference on Empirical Methods in Natural Language Processing. 2022.

**Paper Topic And Main Contributions:**

This paper is about developing a joint language model of speech and text. The authors proposed to learn a language model with both speech and text inputs by concatenating the two representations or alternating them. Specifically, they proposed to use speech units for speech inputs by quantizing features of HuBERT and applying sentencepiece. They also introduce an evaluation metric to measure the cross-modal transfer ability of a trained model. The topic handling in the paper is very interesting but the evaluation seems insufficient.

**Questions For The Authors:**

1) Why using multi-task learning (uLM + tLM) harms performance? For example, Table 4 and Table 5 show that (D) model has lower performances than uLM only (B) model. Since speech and text share linguistic information, it is likely that other modal performance improvements can be made.

2) If we apply the joint language model tasks beyond the cross-modal transferring, is it still CST or AST works better than uLM+tLM?

3) Line #493, about the curse of multilinguality; Speech-only dataset also contains multilingual data (Multilingual Librispeech, Voxpopuli, CommonVoice), hence model (B) seems to be trained multilingual data. What is the exact meaning of lines # 489~495?

4) Can the authors provide AST-only and CST-only results?

5) Which benefits we can earn by training a joint speech and text language model? It seems not well described in the manuscript.

**Reasons To Accept:**

1. The authors explored a timely and interesting topic; developing a unified text and speech language model system.
2. The authors proposed a Context Retrieval Accuracy (CRA) to measure the performance of a cross-modal language model.

**Reasons To Reject:**

1. The proposed CRA appears to lack discernment. Since CRA measures the emission probability of all ground-truth continuation words given prompt words, the probability will be high for any well-trained language models. As the metric shows almost the same results on u2u results, it is wondering if CRA can serve as a reliable evaluation metric.

2. The evaluation is focused on cross-modal transferability only. As the authors proposed a joint language model of speech and text, it seems to be more natural to evaluate the trained model on diverse downstream tasks including speech-only task, text-only task, and speech-text (multimodal) task. The results shown by the authors are only saying that even though we trained a joint language model of audio and speech, the unimodal performance will be degraded and very small cross-modal transferability can be get.

3. The evaluation results of using CST only, AST only, uLM + AST are not shown. Do the CST and AST methods require uLM basically?

4. As the authors pointed out, PELM has limitations for measuring the proposed method. Hence, Table 3 does not give much insight into the benefit of joint language modeling of speech and text. It seems better to explore other downstream tasks instead of using these metrics.

**Reproducibility:**

4: Could mostly reproduce the results, but there may be some variation because of sample variance or minor variations in their interpretation of the protocol or method.

**Reviewer Confidence:**

5: Positive that my evaluation is correct. I read the paper very carefully and I am very familiar with related work.

---

> ### Author Rebuttal · Authors · 2023-08-29
>
> Thank you so much for the feedback.
>
> Re: reject reason (1): Although CRA is almost the same across the u2u cases, it has differences in the u2t and t2u cases for row (E) and row (F) in Table 3. We don’t suggest using CRA as a formal evaluation metric. Instead, we propose it as a proxy for practitioners to understand how well the model can learn to mix the two modalities without actually fine-tuning the LM on a downstream task (similar to perplexity for LMs). As pointed out in the paper, fine-tuning can have multiple variables that can affect the outcome.
> Also, including more modalities in the training data can cause competition between modalities as shown in [Conneau et. al.](https://arxiv.org/pdf/1911.02116.pdf) and [Aghajanyan et al.](https://arxiv.org/pdf/2301.03728.pdf) CRA can serve as a way to measure how well the model learns to connect the two modalities.
>
> Re reject reason (2): We agree that applying the model to more downstream tasks can also be a valuable study. However, our goal is to understand how much transferability we can get between the two modalities for a joint LM, so we choose mainly spoken language understanding tasks, i.e, SLUE, so that we can fine-tune/evaluate on speech/text in all four combinations.
>
> Re reject reason (3): We don’t have the results for CST and AST only. Practically, CST and AST are obtained from paired speech and text data. With such data, we can create uLM and tLM. Also, it is much easier to collect unpaired data in speech or text.
>
> Re reject reason (4): We proposed CRA and PELM as a way to probe the model instead of using it for formal evaluation. We agree that exploring more downstream tasks can also be a great study. We leave it as future work.
>
> Re question (1): Our interpretation is that when the model is only exposed to unimodal speech and text without knowing the connection between the two modalities, it shares its capacity between the two. As a result, the two modalities are more competitive than when using mixed speech-text data. As shown in Fig.5 in [Aghajanyan et al.](https://arxiv.org/pdf/2301.03728.pdf), we need enough scaling to break the competition.
>
> Re question (2):
> From SLUE-NER experiment, we do see an improvement when adding CST and AST (E, F) but not tLM (D). Also, when fine-tuning on text and testing on text (in this case, it becomes a text-only task) on SLUE-SA and SLUE-NER (Table 4), CST and AST also provide improvements.
>
> Re question (3): We'd like to address what seems to be a misunderstanding. Our model was exclusively trained on English data. The concept of the curse of multi-linguility refers to the phenomenon where two types of data (modalities or languages) can compete with each other, as demonstrated in our SLUE-SA experiment. In this experiment, we observed that introducing text data decreased the performance on speech. We will provide further clarification on this aspect in the final version.
>
> Re question (4): We won’t be able to provide these results due to resource constraints. As pointed out in re q(2), uLM and tLM are easier to collect and always come with AST and CST. As a result, we believe that using uLM and tLM along with CST and AST is a reasonable setup.
>
> Re question (5): Potential benefits: As described in the first paragraph of the introduction, adding the two modalities together can have benefits as the data in speech and text have different characteristics (speech: dialog, text: structural knowledge). Our study also shows that mixing the modalities improves transferability for a classification task (SLUE-SA), and improves a text generation task (NER) on speech.

---

### Meta-Review · Area_Chair_GWF4 · 2023-09-14

**Recommendation:** 3

**Metareview:**

This paper looks at combining discrete speech units (automatically extracted with a self-supervised speech model that quantises its input) with text. The proposed approach is applied to spoken language understanding tasks. The reviewers commented positively on the writing of the paper and the thoroughness of the analysis (good soundness scores). But some of the reviewers were concerned about limitations in the experiments, e.g. in the type of downstream tasks used (ambivalent and mediocre excitement).

---

### Decision · Program_Chairs · 2023-10-07

**Decision:**

Accept-Findings

**Comment:**

This paper looks at combining discrete speech units (automatically extracted with a self-supervised speech model that quantises its input) with text. The proposed approach is applied to spoken language understanding tasks. The reviewers commented positively on the writing of the paper and the thoroughness of the analysis (good soundness scores). But some of the reviewers were concerned about limitations in the experiments, e.g. in the type of downstream tasks used (ambivalent and mediocre excitement).